# A Self-Supervised Model Based on CutPaste-Mix for Ductile Cast Iron Pipe Surface Defect Classification

**DOI:** 10.3390/s23198243

**Published:** 2023-10-04

**Authors:** Hanxin Zhang, Qian Sun, Ke Xu

**Affiliations:** Collaborative Innovation Center of Steel Technology, University of Science and Technology Beijing, Beijing 100083, China; m202121256@xs.ustb.edu.cn (H.Z.); d202210616@xs.ustb.edu.cn (Q.S.)

**Keywords:** ductile cast iron pipe, defect classification, self-supervised, CutPaste-Mix

## Abstract

Online surface inspection systems have gradually found applications in industrial settings. However, the manual effort required to sift through a vast amount of data to identify defect images remains costly. This study delves into a self-supervised binary classification algorithm for addressing the task of defect image classification within ductile cast iron pipe (DCIP) images. Leveraging the CutPaste-Mix data augmentation strategy, we combine defect-free data with enhanced data to input into a deep convolutional neural network. Through Gaussian Density Estimation, we compute anomaly scores to achieve the classification of abnormal regions. Our approach has been implemented in real-world scenarios, involving equipment installation, data collection, and experimentation. The results demonstrate the robust performance of our method, in both the DCIP image dataset and practical field application, achieving an impressive 99.5 AUC (Area Under Curve). This presents a cost-effective means of providing data support for subsequent DCIP surface inspection model training.

## 1. Introduction

Defect detection plays a crucial role in industrial production as the timely identification of surface defects is essential for enhancing production quality [1]. A ductile cast iron pipe [2] is a type of pipeline material that can withstand higher pressure and loads and has a longer lifespan and lower maintenance costs. During the production process of Ductile Cast Iron Pipes (DCIPs), various defects [3,4] such as cracks, heavy skin, and pore voids can inevitably arise, impacting product quality and safety. Employing effective defect detection techniques for DCIPs is pivotal in ensuring product longevity and quality. With the advancement of machine vision [5,6] and deep learning [7,8,9], there has been growing interest among numerous research teams in developing online surface defect detection methods [10,11,12]. These systems primarily utilize methods based on two-dimensional feature information. They involve capturing two-dimensional images of the target surface using cameras, analyzing defect information within these images, and subsequently conducting defect classification and detection.

Making an excellent online surface detection system involves many steps. With a sufficient amount of data, existing detection models [13,14] have the potential to reach the state-of-the-art (SOTA) level in particular scenarios. Usually, the accuracy of these methods is positively correlated with the number of defect samples. Thanks to the mature casting process of DCIPs, the production often results in a high yield rate. However, the occurrence of defects is unpredictable both in time and space, making the collection of defect samples a challenging task. We made a simple estimate of the amount of defect data in a production line. Unfortunately, the number of defect images is less than 0.2% of the total collected images. The cost and inefficiency of manually filtering these samples are clearly prohibitive. The production scenario mentioned above poses significant challenges to the collection of defect data. Therefore, our research focuses on efficiently filtering negative samples [15] rather than on proposing a new detection model. This distinction is crucial for understanding the significance of our study.

Constructing an efficient way of collecting defect images is a key factor in the implementation of surface detection. Essentially, the task is defined as anomaly classification, a prerequisite task that supports surface detection. Due to the limited availability of anomaly data, constructing effective anomaly classification models is often achieved through semi-supervised or self-supervised methods. Given the unknown distribution of anomaly regions, training is typically conducted solely on defect-free samples. Thus, the creation of a meaningful pre-training task becomes crucial. Deep One-class [16] has demonstrated an effective end-to-end training model, which is a parameterized deep neural network-based one-class classifier. Notably, its superiority lies in its deeper network architecture, outperforming shallower classification models such as one-class support vector machines or autoencoders [17], as demonstrated through comparative experiments with deep neural networks. In self-supervised feature learning methods, techniques like geometric transformations [18] and contrastive learning [19] have shown success in successfully discriminating normal and defective images.

In this work, we approximately define defect regions as a particular case of image anomalies, wherein a binary classification model should focus on all regions divergent from the normal surface. We adopt a two-stage framework [20], wherein we initially create anomaly regions on normal DCIP surface images through the construction of a self-supervised learning pre-training task. Specifically, we propose a strategy called CutPaste-Mix. This is an improved method based on CutPaste [21], creating abnormal regions by cutting patches from images. CutPaste-Mix includes three strategies, as follows: 1. Not reattaching the patch to the image, thus preserving the incomplete area; 2. Enlarging the patch by a certain factor and then randomly pasting it back onto the image; 3. Randomly rotating the patch and subsequently pasting it back into a random position within the image. Our intention is to introduce irregularities approximating defects in images. Moreover, we utilize ResNet-18 for feature learning on both types of images and employ defect-free feature vectors to compute a Gaussian Density Estimator (GDE) [22,23]. The GDE is leveraged to compute abnormal scores for each image, thereby achieving the classification of anomalous images.

We deployed our equipment on-site and collected actual DCIP images, creating a dataset. We tested our proposed method on the DCIP dataset and compared it with other models. Remarkably, without utilizing any defective data for training, our method achieved an impressive 99.5 AUC (Area Under Curve) [24]. Additionally, we conducted ablation experiments demonstrating the efficacy of the three methods contained within CutPaste-Mix when used individually and in combination. The outcomes demonstrated the effectiveness of the self-supervised classification model based on CutPaste-Mix in anomaly classification. The model we propose proves valuable for dataset creation and holds promise for practical engineering deployment.

## 2. Methods

### 2.1. Definition of Self-Supervised Binary Classifier

A self-supervised binary classifier [25] is a machine learning model designed to learn meaningful representations from unlabeled data by formulating a binary classification task within the data itself. The core idea behind self-supervised binary classification is to create surrogate positive and negative samples from the input data and train the model to differentiate between them. This allows the model to learn useful features from the data without the need for explicit human labeling. Common techniques used to generate positive and negative samples include data augmentation, transformations, or utilizing context information from the same data instance.

In the context of image data, techniques such as data augmentation can be employed to train a self-supervised binary classifier. In this approach, parts of an image are cut and pasted to create positive and negative pairs. The model is then trained to distinguish between the original images and the manipulated ones. This encourages the model to learn relevant features for the given classification task. Mathematically, let x represent an input image, $x^{+}$ represent a positive sample (e.g., an image with a manipulated patch), and $x^{-}$ represent a negative sample (e.g., the original image or a different image). The self-supervised binary classification loss function can be formulated as follows:$$L\left(x,x^{+},x^{-}\right)=-log\left(\frac{exp\left(f\left(x\right)\cdot f\left(x^{+}\right)\right)}{exp\left(f\left(x\right)\cdot f\left(x^{+}\right)\right)+exp\left(f\left(x\right)\cdot f\left(x^{-}\right)\right)}\right)$$
where $L\left(x,x^{+},x^{-}\right)$ represents the loss for a triplet of samples: the input image x, positive sample $x^{+}$, and negative sample $x^{-}$. $f\left(x\right)$ represents the feature embedding of sample x. The formula calculates the binary classification loss using logistic loss (cross-entropy) for positive (x and $x^{+}$) and negative (x and $x^{-}$) pairs.

In summary, a self-supervised binary classifier needs to enable the model to learn informative features from unlabeled data by creating its own binary classification task. This approach taps into the inherent structure of the data to generate supervision signals, making it a powerful technique for learning meaningful representations without the need for extensive manual labeling.

### 2.2. Self-Supervised Learning with CutPaste-Mix

A well-crafted pretext task [26,27] is a cornerstone for achieving successful self-supervised learning. The essence of such a task lies in setting up a specific objective within unlabeled data, enabling the model to glean meaningful representations from it. This task is meticulously designed to generate supervision signals intrinsically from the data itself, thus obviating the need for manual annotations. By tackling this pretext task, the model becomes adept at unraveling the inherent structures and patterns latent within the data in an unsupervised manner. Consequently, it provides a more robust foundation for initializing subsequent supervised learning tasks or refining feature representations. Commonly encountered pretext tasks in the realm of self-supervised learning encompass predicting image rotation angles, completing missing segments, colorizing images, and capturing contextual relationships within images. These tasks harness the intrinsic information embedded within the data to generate supervision signals, thereby prompting the model to assimilate valuable features and elevate its capacity for generalization.

Due to the requirement of training the model exclusively on defect-free data while also formulating an effective pretext task, the utilization of data augmentation techniques to generate images with anomalous regions emerges as the most suitable strategy. This methodology ensures the model’s robust acquisition of distinctive features. While techniques such as rotation, translation, and contrastive learning methods enhance accuracy in single-classification tasks and augment the model’s generalization and resilience against interference, they prove to be less effective when directly applied to images of Ductile Cast Iron Pipes (DCIPs), which are characterized by their high resolution and small defective areas. In essence, augmentation strategies like rotation and translation entail fundamental geometric transformations that are applied uniformly across the entire image. While these strategies are adept at capturing objective and conceptual features (often referred to as semantic features) within images, they fall short in addressing the learning of continuous local features within images. Within the context of DCIP images, defects manifest as irregular and disjointed anomalous regions. Consequently, it becomes imperative to design a strategy that effectively emulates these distinct abnormal regions. In summary, the optimal approach necessitates an augmentation strategy that mimics the irregular and discontinuous nature of DCIP defects, rather than relying solely on generic geometric transformations. This approach will significantly enhance the model’s ability to effectively capture localized anomalies, thereby refining its capability to discern subtle yet crucial features.

We present a self-supervised methodology called CutPaste-Mix, a data augmentation strategy. CutPaste-Mix constitutes a comprehensive augmentation technique encompassing three specific variants: Erase, Enlarge, and Rotate. Although the task may seem straightforward to implement through conventional programming, it is gratifying to note that it indeed empowers deep neural networks to glean distinctive features from normal regions. Fundamentally, CutPaste-Mix involves the random cropping of a patch from a defect-free image and its subsequent overlay onto the original image to simulate an anomalous region.

**CutPaste-Mix.** A patch is randomly cropped from an image of the dataset; then, one or more of the following three strategies (Erase, Enlarge, or Rotate) are randomly used. This process creates anomalous regions on a normal surface, and the processed images serve as positive samples only for training the backbone of the classification network. We also set a consistent random seed to ensure the reproducibility of random processes.

**Erase.** Analogous to the Cutout technique, this strategy directly excises a portion from the image, discarding the patch area. Given that DCIP images are grayscale, the residual area’s color adheres to the original grayscale values, with the mean of the original pixels serving as a substitute color. However, this approach engenders appreciable information loss in images and can be harmonized with other methods.**Enlarge.** Commencing with the random cropping of a rectangular region, this method subsequently enlarges and reattaches it to a random location on the original image. We confine the cropped region’s dimensions to not exceed 20% of the image area, while the enlargement factor remains within the image’s boundaries. This approach often yields significant abnormal regions while mitigating the substantial information loss entailed by erase.**Rotate.** Analogous to Enlarge, this approach involves initial patch cropping, followed by rotation by a random angle before reintegration into the original image. This tactic imposes minimal information loss while concurrently generating abnormal regions.

We chose ResNet-18 as the backbone of the classifier. As shown in Figure 1, the input images serve as negative samples for the training set. Simultaneously, all the output images generated through CutPaste-Mix are positive samples for the training set. It is important to note that the training of the network backbone requires both positive and negative samples. However, when fitting the parameters of the Gaussian density estimator, we exclusively used negative samples. To test whether a new image has anomalies or not, ResNet-18 first computes the feature vector. Then, the Gaussian density estimator calculates the final anomaly score for this feature vector to achieve the classification goal.

This paper introduces the training loss function for the proposed supervised representation learning, as follows:$$L_{CPM}=E_{x\in \chi }\{CE\left(g\left(x\right),0\right)+CE\left(g\left(CP\left(x\right),1\right)\right)$$
where χ represents the defect-free dataset, $L_{CPM}\left(\cdot \right)$ stands for the loss function of the CutPaste-Mix, and $g\left(\cdot \right)$ is the singular classifier composed of a deep neural network.$\;CE\left(\cdot ,\cdot \right)$ denotes the cross-entropy loss. During the implementation of the code, all the data augmentation strategies mentioned in CutPaste-Mix are completed before inputting the sample x into the singular classifier $g\left(\cdot \right)$.

### 2.3. Deep Residual Learning with ResNet-18

We recognize that there are many outstanding Convolutional Neural Networks (CNNs) available for us to choose from as the backbone of our classification model. These CNNs exhibit exceptional feature extraction capabilities. In the production pipeline of DCIPs, the speed of data collection is high. For our proposed classification model to achieve rapid classification on the production line, it must have a lower number of layers and lower computational complexity. Therefore, we excluded models like Inception [28] and Xception [29] due to their higher computational complexity. Traditional backbone networks like VGG [30] and AlexNet [31] sometimes suffer from the vanishing gradient problem because of less effective gradient propagation strategies. We found that ResNet-18′s [32] network complexity and residual strategy align well with our requirements, which is why we chose it as the backbone for constructing the classification model.

Deep Residual Learning, commonly referred to as ResNet, has emerged as a significant advancement in the realm of deep neural networks. One prominent architecture within the ResNet family is ResNet-18 [32,33], which exhibits remarkable capabilities in overcoming the challenges posed by training extremely deep networks. ResNet-18 utilizes skip connections, also known as residual connections, to enable the effective training of very deep networks by mitigating the vanishing gradient problem.

The core innovation of ResNet-18 lies in its residual blocks, where the input to a layer is added to the output of a subsequent layer, allowing for the preservation of features across different layers. Figure 2 showcases the architecture of a residual block. The whole network architecture is composed of several stacked residual blocks, each consisting of multiple convolutional layers, batch normalization, and ReLU activation functions. This enables the network to learn complex hierarchical features while still maintaining a manageable overall network depth. Mathematically, this residual connection can be represented as follows:$$x_{l+1}=x_{l}+F\left(x_{l},W_{l}\right)$$
where $x_{l}$ represents the input to the l-th layer, $x_{l+1}$ is the output of the l+1-th layer, $F\left(x_{l},W_{l}\right)$ is the residual mapping, and $W_{l}$ denotes the weights of the l-th layer.

The ResNet-18 architecture serves as the foundational convolutional backbone in this investigation. Notably, an appended global average pooling layer precedes the terminal fully connected layer within the convolutional hierarchy. The resultant feature representation extracted from this pooling operation serves as the direct input to the ensuing fully connected stratum, which is seamlessly linked to a multilayer perceptron. To facilitate the accommodation of diverse image dimensions and sustain training efficiency invariant to input image size variations, a global average pooling layer is seamlessly integrated prior to the ultimate fully connected stratum of the ResNet-18 framework. Noteworthy is the scenario when images of dimensions 256 × 256 × 3 are channeled into the model, thereby engendering an 8 × 8 × 512 nodal feature map. Notably, the terminal fully connected layer of ResNet-18 encompasses 1000 nodes, necessitating the propagation of 32,768 × 1000 weights, thereby demanding substantial memory resources. The strategic inclusion of the global average pooling layer streamlines the transition from the feature map to the ultimate classification outcome, thereby manifesting substantial empirical efficacy. Concomitantly, this architectural augmentation yields a streamlined parameter space, thus fortifying model resilience and affording a pronounced mitigation of the proclivities toward overfitting.

### 2.4. Computing Anomaly Score for Classification

The realm of binary classifiers offers a diverse array of methodologies for computing anomaly scores. The proposed approach involves the direct computation of anomaly scores based on the features extracted by the convolutional backbone. This is executed through techniques like kernel density estimation [34] or Gaussian density estimation [23]. Over the course of the past several decades, both kernel density estimation and Gaussian density estimation methods have undergone thorough investigation and widespread application. While these methods may exhibit certain limitations in specific contexts, such as parameter selection and computational intricacy, they stand as robust mechanisms for estimating probability density functions. These approaches offer invaluable tools and insights for tasks such as anomaly region classification. In our study, we harness the Gaussian density estimation technique to calculate anomaly scores for anomalies present in DCIPs, thereby facilitating an effective method for classifying surface defects in railway tracks.

Kernel Density Estimation offers several advantages, including the absence of prior assumptions about data distribution and the avoidance of the necessity for pre-estimation of parameters. We construct a simple parametric Gaussian density estimator, the basic principle of which is expressed mathematically as follows:$$\log p_{\mathrm{gde}\;}\left(x\right)\propto \left\{-\frac{1}{2}{\left(f\left(x\right)-\mu \right)}^{⊤}\Sigma^{-1}\left(f\left(x\right)-\mu \right)\right\}$$
where μ and Σ represent the Gaussian parameters learned from defect-free negative samples.

## 3. Experiments

### 3.1. Description of DCIPs

The experiment focuses on inspecting T-type centrifugal ductile cast iron pipes (DCIPs), as illustrated in Figure 3. These specific pipes play crucial roles in municipal, industrial, and mining water supply and drainage systems, as well as contributing to rural water supply networks. Their significance extends to enhancing both urban and rural water supply conditions and augmenting regional sources of rural drinking water. Hence, it becomes imperative to ensure the quality and performance of cast pipes through thorough and dependable testing before they are dispatched from the factory.

The practical application scenarios vary, giving rise to diverse prerequisites concerning DCIP models, sizes, and various parameters. The specifications of the T-type centrifugal ductile cast iron pipes intended for assessment on the production line are elucidated in Table 1. The dimensions of DCIPs on the production line exhibit variations, encompassing a range from DN350 to DN1000 mm, with a maximum nominal diameter deviation of 650 mm. Given the cylindrical nature of the DCIP, rotational motion is indispensable for effectively detecting the circumferential surface. Moreover, these cast pipes extend up to a length of 6 m, necessitating the deployment of 6 cameras to cover the entire span, even though a single camera can encompass a physical field of view of 1 m.

### 3.2. Experiment Setup

To address the challenge posed by variations in the nominal diameters of Ductile Cast Iron Pipes (DCIPs), we introduce an automated lifting adjustment system predicated on Programmable Logic Controllers (PLCs) and servo motor technologies. This system operates by discerning the specific diameter of the targeted DCIP for inspection and subsequently orchestrating the repositioning of the vision system to an optimized focal length for precise image acquisition. This real-time dynamic adjustment ensures the preservation of high-fidelity image capture, which in turn serves as a foundational cornerstone for subsequent algorithmic processing. As delineated in Figure 4, the multistep defect detection procedure for DCIPs within the production environment involves the following pivotal phases:

Step 1: Upon the DCIP’s spatial alignment with the designated detection position, the automated lifting and adjustment system conducts a precise dimensional assessment of the DCIP, subsequently dictating the precise positioning of the imaging module for optimal acquisition.

Step 2: Following the precise alignment of the imaging module with the designated acquisition position, a rotational mechanism is actuated to induce the DCIP’s rotation. This rotational mechanism maintains a consistent angular velocity despite varying linear speeds resultant from inherent DCIP size discrepancies. The synchronization of the rotational velocity with the light source system and camera acquisition is accomplished through encoder feedback, enabling real-time modulation of the light source’s luminance duration and the camera’s exposure time. Each imaging module seamlessly transfers acquired image data to a centralized server.

Figure 5a,b provides an illustrative depiction of the prototype system’s configuration seamlessly integrated within the production line, emanating from the foundational prototype framework aforementioned. The illumination infrastructure within this context is meticulously orchestrated by two distinct lighting controllers, both intricately interfaced with a computational unit. Employing high-resolution BASLER Mono CMOS line scan cameras (Ahrensburg, Germany) boasting 4096 pixels, accompanied by ZEISS Planar T* 1.4/50 mm lenses, the image capture process is orchestrated. These cameras are intricately synchronized with the LED lighting infrastructure through a Field-Programmable Gate Array (FPGA) control board, ensuring temporal harmony. The captured DCIP images are promptly relayed to the computational unit for further comprehensive processing. The computational infrastructure boasts a 64-bit Windows 10 operating system, a 2.4 GHz central processing unit, and a substantial 32 GB of Random Access Memory (RAM).

### 3.3. Description of Image in Dataset

The imaging assessment apparatus deployed at the production site was employed for image testing on each machined DCIP. The system utilized a line scan camera with a resolution of 4096 pixels. By configuring the image acquisition parameters through the Pylon software, 4096 rows of data were simultaneously captured, resulting in a DCIP image with dimensions of 4096 × 2048 pixels. An exemplar image of the DCIP sample acquired on the production line is depicted in Figure 6.

There are six common types of surface defects found on DCIP: crazing, cracks, heavy skin, iron bean, mold powder, and pores. The visual characteristics of these defects are presented in Figure 7. After segmenting the captured 4096 × 2048-pixel images into consecutive 512 × 512-pixel sections and applying subsequent filtering, the obtained surface images devoid of any anomalies are shown in Figure 8. The captured images with dimensions of 4096 × 2048 pixels were segmented into consecutive images of size 512 × 512 pixels.

### 3.4. Training and Testing Process of Classifier

**Training Details:** Our experiments were conducted on an NVIDIA RTX 3090. The input images were of size 512 × 512 pixels. The model typically converges with 150 iterations. We also utilized a list-type hyperparameter, [‘erase’, ‘enlarge’, ‘rotate’], to control the involvement of the three strategies in CutPaste-Mix. The optimizer used was SGD (Stochastic Gradient Descent) with a learning rate of 0.03, a momentum parameter of 0.9, and a weight decay parameter set to $3\times 10^{-5}$.

**Dataset Preparation:** Utilizing the aforementioned acquisition equipment, images of DCIPs were collected. Around 2000 defect-free images were meticulously selected to serve as the negative samples for the training set. These images were subsequently cropped into a 512 × 512-pixel format, as illustrated in Figure 8. The dataset was then subjected to CutPaste-Mix to generate images containing anomalous regions, thereby constituting the positive samples for the training set. Within the employed training dataset, an equal number of positive and negative samples were maintained.

The training steps of combining a pre-trained deep convolutional network with Gaussian density estimation to create a binary classifier can be represented as follows:

**Computation of Feature Vectors:** Employing a pre-trained ResNet-18, the training dataset undergoes feature extraction. Each datum is fed through the initial layers of the convolutional neural network, thus obtaining the image’s feature representation. These representations can be conceived as high-dimensional feature vectors of the images.

**Computing the parameters of GDE:** The mean and covariance matrix of the Gaussian density estimator are computed using defect-free DCIP data. For each positive and negative sample, we aim to set an appropriate threshold to distinguish between them. Thus, we use the testing set to iteratively adjust suitable thresholds. This threshold is continuously optimized as new defect data are gathered in subsequent iterations.

**New Samples classification:** The images are first passed through the pre-trained convolutional neural network to obtain the feature representation of new samples. Next, these feature representations are input into the Gaussian Density Estimator (GDE) model to calculate the likelihood probabilities of the sample belonging to the positive and negative classes. Finally, the class with the higher probability is chosen as the prediction result.

**Accuracy evaluation:** We used 200 real samples with defects as the testing set, all of which were collected using the device proposed in this paper. The AUC (Area Under Curve) is the evaluation metric used to test the performance of our classifier and other models. The classifier calculates the True Positive Rate (TPR) and False Positive Rate (FPR) using various thresholds on the test dataset and plots these values to create the ROC curve. The ROC curve typically has TPR on the *y*-axis and FPR on the *x*-axis. Finally, the area under ROC curve, denoted as AUC (Area Under the Curve), is calculated to evaluate the classifier’s performance.

### 3.5. Main Results

Table 2 showcases the performance of the novel self-supervised classification approach introduced in this study. Through 10 distinct experiments employing diverse random seeds, we present the averaged AUC accompanied by its standard error across the testing set. We not only evaluated the overall performance of CutPaste-Mix but also separately evaluated the three augmentation strategies it includes: Erase, Enlarge, and Rotate. Additionally, we conducted comparative assessments with three alternative techniques: Deep One-Class (DOCC) [16], Uninformed-Student [35], and Patch-SVDD [36].

We refer to the three strategies included in CutPaste-Mix as Enlarge, Erase, and Rotate just for clarity. Enlarge excelled as a standalone data augmentation technique, even surpassing Rotate, achieving an impressive 95.8 AUC. The performance was poorest while only using Erase, with 84.3 AUC. When combining two data augmentation methods, Enlarge and Rotate together achieved an even higher 98.9 AUC. Furthermore, the results of CutPaste-Mix reached a remarkable 99.4 AUC. We also compared the top-performing CutPaste-Mix approach with other models, and it consistently yielded the best results, as demonstrated in Table 3.

We applied our most effective classifier to the field-collected dataset using the acquisition equipment. The results were gathered from 10,000 consecutive screening operations. Our proposed self-supervised classifier ultimately identified 43 images as anomalous. Following on-site comparisons, we validated the presence of 20 genuine defect images, as illustrated in Figure 9. However, it is important to note that the DCIP’s surface is inevitably subject to disturbances like water stains and oil smudges, as depicted in Figure 10. These noise interferences also contribute to the anomalous regions. While it may not be feasible to entirely distinguish whether the detected anomalous regions represent defects or noise, our approach substantially reduced the time required for manual defect screening. Moreover, its on-site applicability offers significant convenience for subsequent model optimization endeavors.

## 4. Conclusions

The aforementioned experimental results confirm the feasibility and superiority of the self-supervised classification algorithm based on CutPaste-Mix on the DCIP dataset. The use of data augmentation to generate abnormal regions in images effectively yields positive samples for classifier training. The ResNet-18 backbone network efficiently captures image features and computes feature vectors. The Gaussian Density Estimation (GDE) is utilized to compute anomaly scores for the extracted features, thus achieving anomaly classification. Our experimental outcomes demonstrate the advantages of this approach compared to other methods. Through ablation experiments, we discussed the results of using different augmentation strategies individually and in combination. For each method, we conducted 10 trials and calculated the average AUC and standard deviation to showcase the superiority of CutPaste-Mix.

This study also has a limitation. In the classification results, we encountered non-defective surface images, such as water stains or oil stains. This suggests that the proposed classification model struggles to distinguish between genuine defects and noise interference, both of which fall into the category of anomalies. The reason for this challenge may be that both genuine defects and noise interference have a large Euclidean distance from normal surfaces in the feature space, but their feature similarity is high. Therefore, we believe that distinguishing between noise interference and real defects could be a valuable direction for future research.

## Figures and Tables

**Figure 1 sensors-23-08243-f001:**
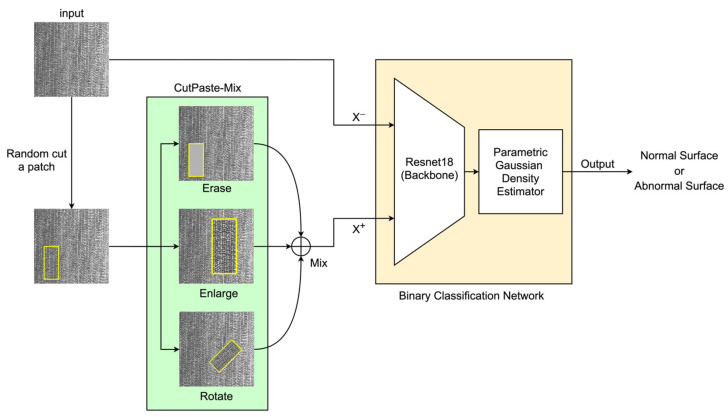
CutPaste-Mix includes three augmentation methods (Erase, Enlarge, Rotate) for creating abnormal regions on a normal surface. The binary classification network consists of ResNet-18 (backbone) and a parametric Gaussian density evaluator. “Normal Surface” refers to a surface that is within the expected or acceptable condition, and “Abnormal Surface” is the opposite.

**Figure 2 sensors-23-08243-f002:**
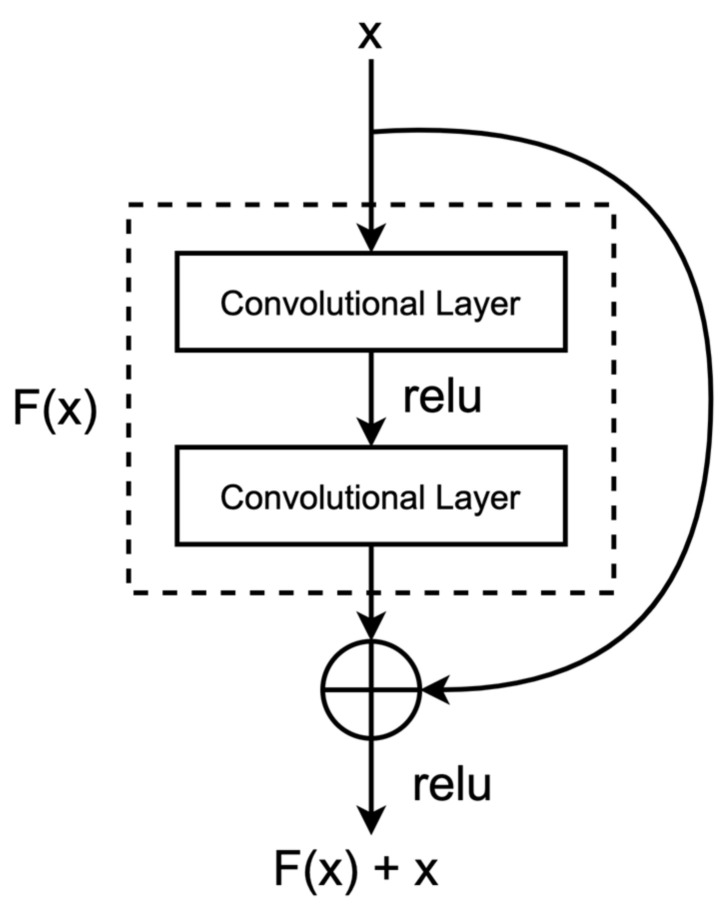
In a residual structure, the input features undergo initial processing through a sequence of convolutional layers and activation functions. These processed features are combined with the output through a skip connection, achieved by element-wise addition.

**Figure 3 sensors-23-08243-f003:**
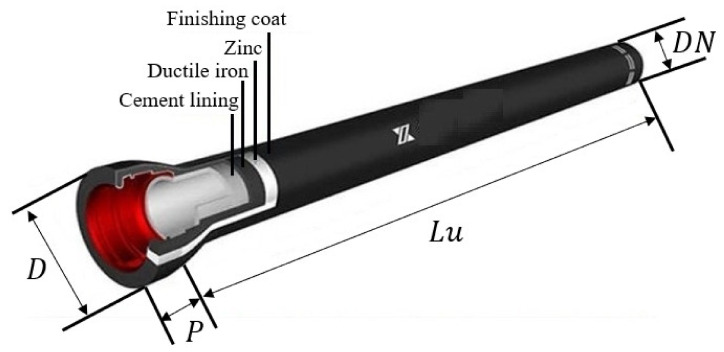
The T-type centrifugal DCIP.

**Figure 4 sensors-23-08243-f004:**
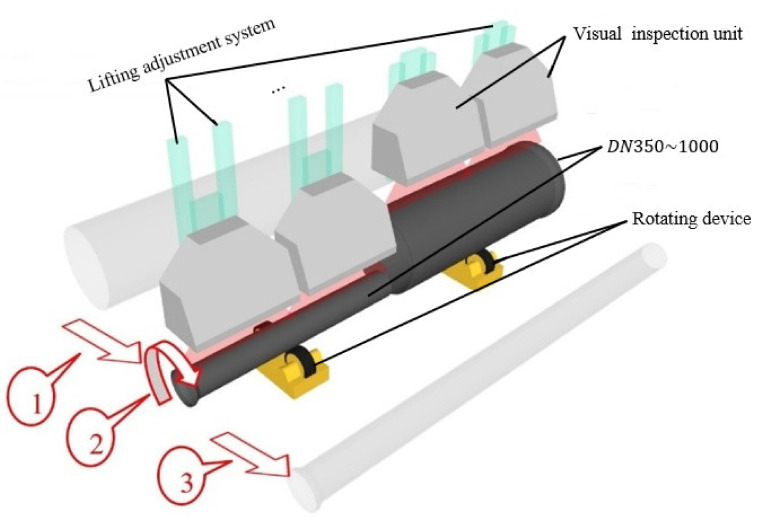
The prototype system of capture DCIP data.

**Figure 5 sensors-23-08243-f005:**
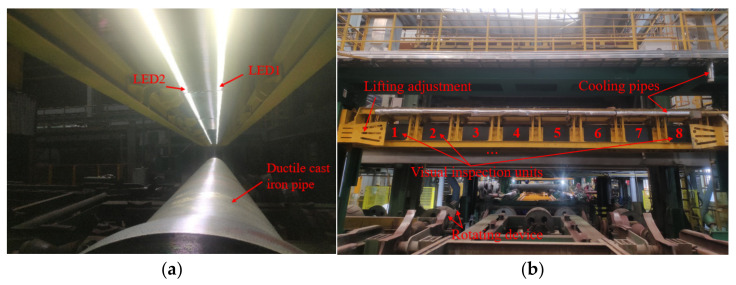
The experimental prototype system at the production site. (**a**) Local scene; (**b**) global scene.

**Figure 6 sensors-23-08243-f006:**
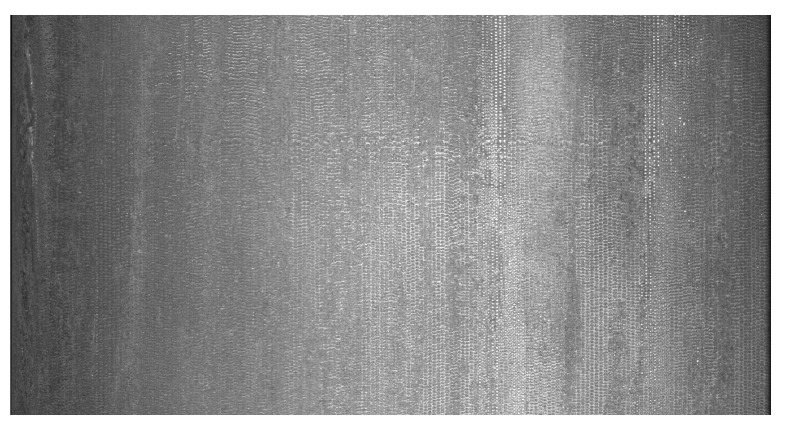
Sample images of DCIP (uncut).

**Figure 7 sensors-23-08243-f007:**
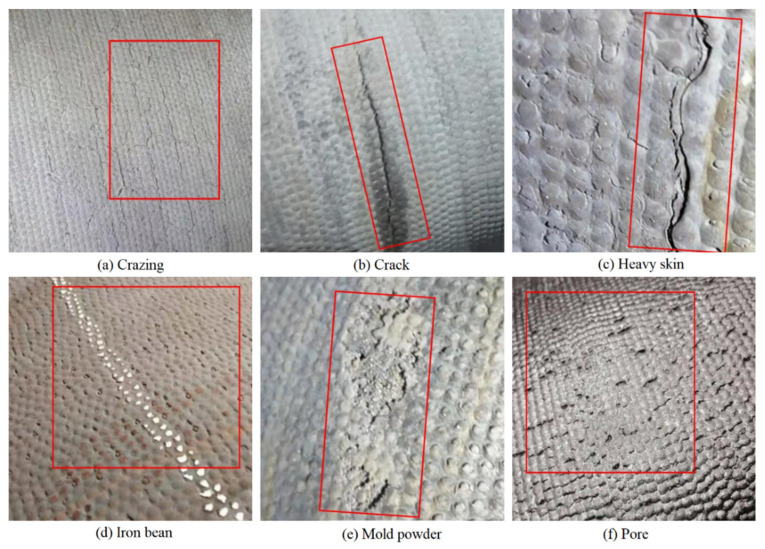
Images of DCIP with real surface defects, (**a**–**f**) in turn expressed as crazing, crack, heavy skin, iron bean, mold powder, pore.

**Figure 8 sensors-23-08243-f008:**
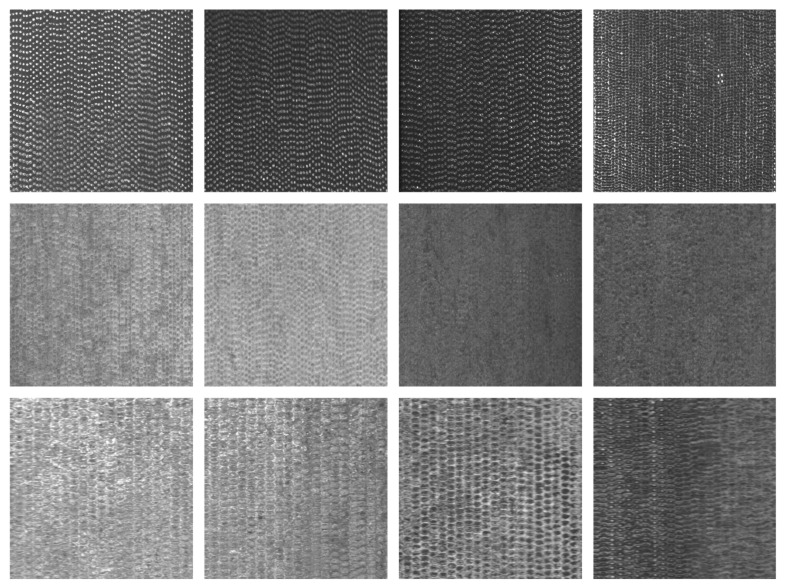
Normal defect-free surface of DCIP.

**Figure 9 sensors-23-08243-f009:**
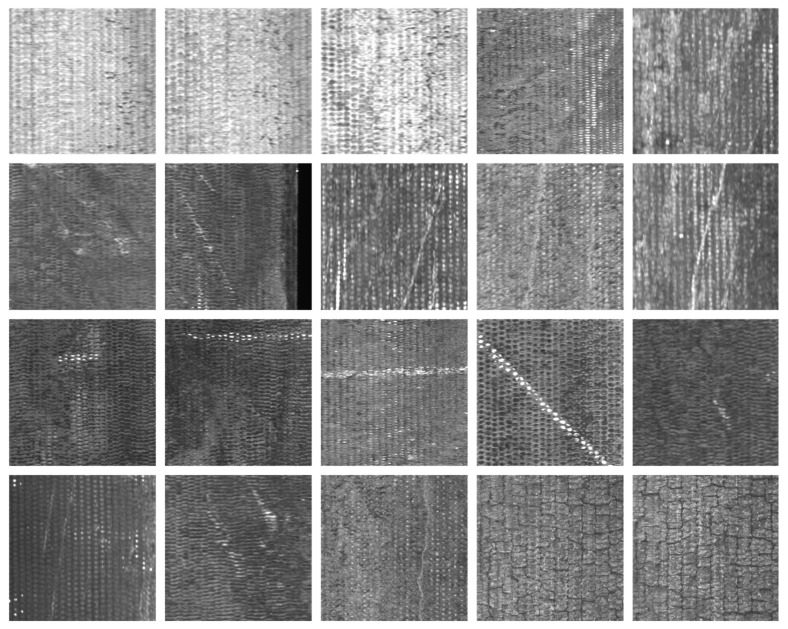
The true positives in the model’s classification results encompass defect types such as iron bean, crack, crazing, and pore.

**Figure 10 sensors-23-08243-f010:**
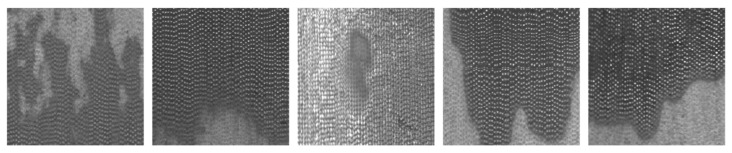
The false positives in the model’s classification results are primarily caused by oil stains, water marks, and iron oxide patches on the surface of the DCIPs. Although these anomalies are not defects, they still represent abnormal areas distinct from the normal surface.

**Table 1 sensors-23-08243-t001:** The specifications of the T-type centrifugal DCIPs.

DN/mm	D/mm	P/mm	Lu/m
350	448	110	6
400	500	110	6
450	540	120	6
500	604	120	6
600	713	120	6
700	824	150	6
800	943	160	6
900	1052	175	6
1000	1158	185	6

**Table 2 sensors-23-08243-t002:** The ablation experiments evaluated the performance of CutPaste-Mix and its sub-methods separately on the DCIP dataset. To reduce randomness, each experimental group was tested 10 times using different random seeds. We report the AUC and standard deviation for each experimental group.

Erase	Enlarge	Rotate	AUC
*			84.3 ± 2.3
	*		95.8 ± 1.1
		*	92.5 ± 1.8
*	*		88.6 ± 0.5
*		*	97.2 ± 0.7
	*	*	98.9 ± 0.2
*	*		94.8 ± 1.5
*	*	*	**99.4 ± 0.1**

**Table 3 sensors-23-08243-t003:** The performance of various anomaly detection models on the DCIP dataset evaluated using the AUC metric, serving as a comparative experiment against the CutPaste-Mix.

DOCC [16]	U-Student [35]	P-SVDD [36]	CutPaste-Mix (Best)
83.4	92.8	95.6	**99.5**

## Data Availability

The data that support the findings of this study are available from the corresponding author upon reasonable request.

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
