# Peer review of "A Self-Supervised Model Based on CutPaste-Mix for Ductile Cast Iron Pipe Surface Defect Classification"

_sensors, 2023, doi:10.3390/s23198243_

Round 1

Reviewer 1 Report

Surface defect monitoring is quite important to ensure the safe operation of pipeline. This paper proposes a CutPaste-based approach for DCIP surface defect monitoring. The research is interesting, however, the following comment should be deal with before acceptance.

i. Please check the writing, as I can see some gramma mistakes.

ii. CutPaste is not a novel approach, and there are also papers try to use this approach for pipeline surface defect monitoring. So what is the contributions of the paper compared with existing approach? It should be more clear.

iii. What are the advantages of this approach cpmpared with other online surface inspection approach?

iv. Please compare the approach with advanced machine vision approach.

Author Response

Dear reviewers:

Thanks very much for your careful review and positive comments. We have substantially revised our manuscript sensors-2601049 entitled 'A Self-Supervised Model based on CutPaste for Ductile Cast Iron Pipes Surface Defect Classification', after reading the constructive suggestions provided by the three reviewers.

Please note that we have replaced 'CutPaste' in the title with a more accurate term, 'CutPaste-Mix.' So, the new title is 'A Self-Supervised Model based on CutPaste-Mix for Ductile Cast Iron Pipes Surface Defect Classification'. We believe this change is necessary to express our approach more clearly to readers.

In addition to the suggestions provided by the reviewers, we have also made some necessary revisions to the formulas and text to better clarify our research content. We have highlighted the revisions for your clear review. Certification for the clear and separate indications of our modifications is attached as the following. Please feel free to contact us with any questions and we are looking forward to your consideration.

Comment1: Please check the writing, as I can see some gramma mistakes.

ANSWER: Thank you for point that out, we have carefully checked and corrected grammatical mistakes.

Comment2: CutPaste is not a novel approach, and there are also papers try to use this approach for pipeline surface defect monitoring. So, what is the contributions of the paper compared with existing approach? It should be clearer.

ANSWER: We agree that CutPaste is not a novel method. However, our research goal is to apply the CutPaste-Mix (an improved CutPaste strategy) to the task of anomaly classification, rather than directly conducting defect detection. We have reviewed relevant literature and have not found a method or scenario that is entirely identical to ours. 

Therefore, we have added relevant explanations and justifications in the article. For details, please refer to highlighted text in the second and fourth paragraphs of Introduction. In section 2.2, we provide a clearer and more detailed explanation of the specifics of our method.

Comment3: What are the advantages of this approach compared with other online surface inspection approach?

ANSWER: We built a binary classification model in order to better collect the defect data, the model can be trained using only normal surface data. It’s important to note that our research focus is on anomaly classification, which constitutes a distinct research domain from defect detection. Therefore, we believe that it's not necessary to conflate them. We believe that some expressions may have caused misunderstanding, and deeply apologize for that.

To provide clarity on this distinction, we have sketched the roles and positions of anomaly classification and defect detection within online surface detection systems. For more details, please refer to the second paragraph of Introduction and Page2, line 49.

Comment4: Please compare the approach with advanced machine vision approach.

ANSWER: In the experiment, we have already compared CutPaste-Mix with highly regarded binary classification models: Deep One-Class (DOCC), Uniformed-Student, and Patch-SVDD. 

Reviewer 2 Report

The authors address the problem of online surface inspection system particularly the fact that the assessment of gathered images would demand a significant amount of time. They propose a self-supervised binary classification algorithm to recognize images of ductile cast iron pipes (DCIP) showing damages based on machine vision and deep learning. The authors construct their own anomaly classification models, and achieve good results. 

The paper is well written and structured. The authors employed modern techniques. Hence, the research work and the achieved results have some good merits for publication. There are only few, mostly technical aspects to be addressed: 

1) All used abbreviations should be introduced when used for the first time and this is valid even if the authors believe that some abbreviations are well-known (for instance, AUC). 

2) When using references, do not put them next to the words, but an empty space between them. Hence, instead of "...quality[1].", write "...quality [1]." 

3) Be consistent when writing words. In the beginning, you wrote CutPaste, later Cutpaste. 

4) When referring to figures, do not put a dot after the figure number. Hence, instead of "Figure 2.", write "Figure 2". ("Figure 2. showcases...")

5) The lines after equations starting with "where" should not be indented (this is not a new paragraph) and should actually start with a small letter (as it is continuation of the same sentence). Alternatively, if it is a new sentence, then start with "Here, ..."

6) Figure 9 is on page 13, but the authors refer to it on page 14 for the first time. A figure should appear only after it was referred to in the text for the first time. 

7) Again, consistency in writing is important. Line 382: "...the results reached a remarkable 99.4 AUC." Number 99.4 is written partly italic, partly by using normal font. Only normal font should be used, as it is the case with all other numbers. 

8) Conclusions should better emphasize the limitations of the proposed technique. They are partly mentioned as an incapability of the model to entirely distinguish between noise interferences and actual defects, but this should be given a bit more attention. 

Author Response

Dear reviewers:

Thanks very much for your careful review and positive comments. We have substantially revised our manuscript sensors-2601049 entitled 'A Self-Supervised Model based on CutPaste for Ductile Cast Iron Pipes Surface Defect Classification', after reading the constructive suggestions provided by the three reviewers.

Please note that we have replaced 'CutPaste' in the title with a more accurate term, 'CutPaste-Mix.' So, the new title is 'A Self-Supervised Model based on CutPaste-Mix for Ductile Cast Iron Pipes Surface Defect Classification'. We believe this change is necessary to express our approach more clearly to readers.

In addition to the suggestions provided by the reviewers, we have also made some necessary revisions to the formulas and text to better clarify our research content. We have highlighted the revisions for your clear review. Certification for the clear and separate indications of our modifications is attached as the following. Please feel free to contact us with any questions and we are looking forward to your consideration.

Comment1: All used abbreviations should be introduced when used for the first time and this is valid even if the authors believe that some abbreviations are well-known (for instance, AUC). 

ANSWER: Thank you for point that out, we have made the revisions to the manuscript according to your suggestions. We have highlighted the revisions for your clear review.

Comment2: When using references, do not put them next to the words, but an empty space between them. Hence, instead of "...quality[1].", write "...quality [1]." 

ANSWER: Thank you for point that out, we have made the revisions to the manuscript according to your suggestions. We have highlighted the revisions for your clear review.

Comment3:  Be consistent when writing words. In the beginning, you wrote CutPaste, later Cutpaste. 

ANSWER: Thank you for point that out, we have made the revisions to the manuscript according to your suggestions. We have highlighted the revisions for your clear review.

Comment4: When referring to figures, do not put a dot after the figure number. Hence, instead of "Figure 2.", write "Figure 2". ("Figure 2. showcases...")

ANSWER: Thank you for point that out, we have made the revisions to the manuscript according to your suggestions. We have highlighted the revisions for your clear review.

Comment5: The lines after equations starting with "where" should not be indented (this is not a new paragraph) and should actually start with a small letter (as it is continuation of the same sentence). Alternatively, if it is a new sentence, then start with "Here, ...".

ANSWER: Thank you for point that out, we have made the revisions to the manuscript according to your suggestions. We have highlighted the revisions for your clear review.

Comment6:  Figure 9 is on page 13, but the authors refer to it on page 14 for the first time. A figure should appear only after it was referred to in the text for the first time. 

ANSWER: Thank you for point that out, we have made the revisions to the manuscript according to your suggestions. We have highlighted the revisions for your clear review.

Comment7: Again, consistency in writing is important. Line 382: "...the results reached a remarkable 99.4 AUC." Number 99.4 is written partly italic, partly by using normal font. Only normal font should be used, as it is the case with all other numbers. 

ANSWER: Thank you for point that out, we have made the revisions to the manuscript according to your suggestions. We have highlighted the revisions for your clear review.

Comment8: Conclusions should better emphasize the limitations of the proposed technique. They are partly mentioned as an incapability of the model to entirely distinguish between noise interferences and actual defects, but this should be given a bit more attention. 

ANSWER: We have made modifications to the conclusion and highlighted them for your review. For more details, please refer to the Conclusion.

Reviewer 3 Report

The current manuscript has some novelty in proposed contribution. The experimental results provide fair comparison. It needs revision in terms of technical details before acceptance. Some comments are suggested.

1. Why did you use ResNet18? There are many other CNNs which can be used as deep neural network in the proposed method. Discuss about the reasons.

2. How do you evaluate the accuracy? Did you calculate the true positive and false positive in size of pixel or window size? Did you use grand-truth images to evaluate the performance?   

3. It is necessary to discuss about the hyper parameter optimization of the used deep network.

4. Related works is a main section in scientific papers. The main scope of this paper is surface defect detection. For example, I find a paper titled “Fabric defect detection based on completed local quartet patterns and majority decision algorithm”, which has enough relation. Cite this paper and some other papers as related works.

5. It is suggested to compare the performance of your proposed approach with some other methods in this scope. Some of general surface defect detection approaches can be applied to DCIP.

6. Shared process should be discussed in a clear way (figure 1).

7. It is suggested to show the main steps of your proposed approach in block diagram format.

Author Response

Dear reviewers: 

Thanks very much for your careful review and positive comments. We have substantially revised our manuscript sensors-2601049 entitled 'A Self-Supervised Model based on CutPaste for Ductile Cast Iron Pipes Surface Defect Classification', after reading the constructive suggestions provided by the three reviewers. 

Please note that we have replaced 'CutPaste' in the title with a more accurate term, 'CutPaste-Mix.' So, the new title is 'A Self-Supervised Model based on CutPaste-Mix for Ductile Cast Iron Pipes Surface Defect Classification'. We believe this change is necessary to express our approach more clearly to readers. 

In addition to the suggestions provided by the reviewers, we have also made some necessary revisions to the formulas and text to better clarify our research content. We have highlighted the revisions for your clear review. Certification for the clear and separate indications of our modifications is attached as the following. Please feel free to contact us with any questions and we are looking forward to your consideration.

Comment1: Why did you use ResNet18? There are many other CNNs which can be used as deep neural network in the proposed method. Discuss about the reasons.

ANSWER: Thank you for point that out. In the first paragraph of section 2.3, we have added a discussion regarding the reasons for choosing Resnet-18 as the backbone of our model, and have highlighted the text.

Comment2: How do you evaluate the accuracy? Did you calculate the true positive and false positive in size of pixel or window size? Did you use grand-truth images to evaluate the performance?   

ANSWER: Thank you for point that out. In our evaluation, we calculate accuracy using metrics such as the Area Under Curve (AUC). True Positive Rate (TPR) and False Positive Rate (FPR) are calculated at the pixel level. We used ground-truth images to evaluate performance. 

We have added a subsection in section 3.4, titled "Accuracy Evaluation," to explain the evaluation process and the quantity of the testing set. We have highlighted the modified text for your review.

Comment3: It is necessary to discuss about the hyper parameter optimization of the used deep network.

ANSWER: Thank you for point that out. We have added a subsection in section 3.4, titled "Training Details." In this subsection, we provide information on some necessary hyperparameters, and the type of GPU. We have highlighted the modified text for your review.

Comment4: Related works is a main section in scientific papers. The main scope of this paper is surface defect detection. For example, I find a paper titled “Fabric defect detection based on completed local quartet patterns and majority decision algorithm”, which has enough relation. Cite this paper and some other papers as related works.

ANSWER: Thank you for point that out. We have already added this paper to the references as per your request and cited it in the Introduction. Please refer to Page 1, Line 36 for details.

Comment5: It is suggested to compare the performance of your proposed approach with some other methods in this scope. Some of general surface defect detection approaches can be applied to DCIP.

ANSWER: Thank you for point that out. In the experiment, we have already compared CutPaste-Mix with highly regarded binary classification models: Deep One-Class (DOCC), Uniformed-Student, and Patch-SVDD. 

It’s important to note that our research focus is on anomaly classification, which constitutes a distinct research domain from defect detection. Therefore, we believe that it's not necessary to conflate them. We built a binary classification model in order to better collect the defect data, the model can be trained using only normal surface data. We believe that some expressions may have caused misunderstanding, and deeply apologize for that. 

To provide clarity on this distinction, we have sketched the roles and positions of anomaly classification and defect detection within online surface detection systems. For more details, please refer to the second paragraph of Introduction and Page2, line 49. In section 3.5, we have also made some modifications to the text, and the modified sections have been highlighted.

Comment6: Shared process should be discussed in a clear way (figure 1).

ANSWER: Thank you for point that out. We directly made modifications to the Figure 1, showing more process details. We also added relevant textual explanations. The detailed content begins from Page 5, Line 176.

Comment7:  It is suggested to show the main steps of your proposed approach in block diagram format.

ANSWER: Thank you for point that out. The answer is the same as in response to Question 6. Now Figure 1 and relevant texts provide a more detailed overview of the method's workflow. The detailed content begins from Page 5, Line 176. 

In Section 3.4, We have also provided a more detailed account of the training and testing steps, and these modifications have been highlighted for your review.

Round 2

Reviewer 1 Report

I recommend to publish this manuscript

Reviewer 3 Report

Most of comments have been considered by the authors. The revised version is better than original submission in methodology details. No more comments are suggested.